

# Heatstroke-induced hepatocyte exosomes promote liver injury by activating the NOD-like receptor signaling pathway in mice

Yue Li[1,2], Xintao Zhu[1,2], Ming Zhang[3], Huasheng Tong[1,2] and Lei Su[1,2]

[1] Department of Intensive Care Unit, General Hospital of Southern Theatre Command of PLA, Guangzhou, China
[2] Key Laboratory of Hot Zone Trauma Care and Tissue Repair of PLA, General Hospital of Southern Theatre Command of PLA, Guangzhou, China
[3] Department of Intensive Care Unit, The Sixth Affiliated Hospital of Guangzhou Medical University, Qingyuan People's Hospital, Qingyuan, China

Corresponding authors
Huasheng Tong, fimmuths@163.com
Lei Su, slei_ccm@163.com

## ABSTRACT

**Background**. Liver injury is a common and important clinical issue of severe heat stress (HS), which has toxic effects and promotes subsequent multiple organ failure. The pathogenesis of HS-induced liver injury has not been fully elucidated. Passively injured hepatocytes also drive liver injury. Exosomes, extracellular vesicles secreted by hepatocytes as "danger signals," mediate the intercellular transportation of diverse functional protein cargoes and modulate the biological processes of target cells. However, whether hepatocyte exosomes are involved in HS-induced liver injury has not been reported. The purpose of the current study was to clarify the release of hepatocyte exosomes under HS conditions and to explore their role in mediating HS-induced liver injury.

**Methods**. HS was induced in hepatocytes or mice by hyperthermic treatment at 43.0 °C for 1 h. Exosomes from control and HS-exposed hepatocytes were isolated by standard differential ultracentrifugation. The hepatocyte exosomes were characterized, and the differentially expressed proteins of the control and HS exosomes were identified by isobaric tags for relative and absolute quantitation (iTRAQ) mass spectrometry and subjected to Kyoto encyclopedia of genes and genomes (KEGG) pathway analysis. Recipient hepatocytes were treated with control or HS exosomes, whereas in vivo, the exosomes were infused into mice. The internalization of HS hepatocyte exosomes by hepatocytes or the liver was tracked. The effect of HS exosomes on the activation of the NOD-like receptor signaling pathway and liver injury was demonstrated in vitro and in vivo.

**Results**. HS induced an increase in the release of exosomes from hepatocytes, which were internalized by recipient liver cells in vitro and taken up by the liver in vivo. HS significantly changed the proteomic profiles of hepatocyte exosomes based on the iTRAQ analysis. The KEGG pathway analysis revealed the enrichment of proteins associated with injury and inflammatory signaling pathways, especially the NOD-like receptor signaling pathway, the activity of which was upregulated. Subsequently, the capacity of HS hepatocyte exosomes to activate the NOD-like receptor signaling pathway was verified and found to aggrevate liver damage and inflammation in vitro and in vivo.

**Conclusions**. This study is the first preliminary study to demonstrate the induction of acute liver injury by hepatic exosomes in the setting of severe HS and reveals potentially related pathways. These results provide a basis for future research and the identification of new targets for clinical intervention.

## INTRODUCTION

Heatstress (HS), with a shared finding of extreme hyperthermia (i.e., a rise in core body temperature >40.5 °C), is the most hazardous illnesses that usually occurs during intensive exercise or exposure to high environmental temperatures (*Epstein & Yanovich, 2019*). Early rapid and effective alleviation of hyperthermia is the cornerstone of treatment; however, it may not fully prevent secondary multiple organ failure (MOF), including acute hepatic failure (AHF), acute renal failure, rhabdomyolysis, disseminated intravascular coagulation (DIC), which eventually causes a high mortality rate (exceeding 35%) (*Hifumi et al., 2018*).

Acute liver injury (ALI) is a key clinical feature of HS. While mild and moderate hepatic injury is relatively common in HS (*Glazer, 2005*), few patients undergo fatal extensive liver failure (*Hassanein et al., 1992*; *Ichai et al., 1997*). More importantly, the injured liver may become an important promoter of late-stage MOF (*European Association for the Study of the Liver, 2017*). However, the pathogenesis of severe HS-related ALI is still unclear; current studies suggest that it cannot be thoroughly explained by direct cytotoxicity but rather systemic inflammatory response syndrome (SIRS) triggered by primary high-thermal injury (*Lim, 2018*).

Exosomes are extracellular vesicles that mediate the transportation of a variety of biologically active molecules, such as proteins, lipids, mRNAs and microRNAs, among others, which regulate target cell function and are part of a novel intercellular signaling system (*Yuana, Sturk & Nieuwl, 2013*; *Bebelman et al., 2018*). As their cargoes are highly disease- and cell type-specific and highly stable due to the protection of the membrane structure, exosomes are an advantageous method of cell-to-cell communication (*Masyuk, Masyuk & LaRusso, 2013*). Hepatocytes are the main exosome-secreting cells in the liver and are relatively vulnerable to various insults. Moreover, hepatocytes are not simply passive targets of injury; they also actively participate in liver damage by releasing "danger signals" (*Pisetsky, 2014*). Exosomes are a type of damage-associated molecular pattern (DAMP) released by hepatocytes after stress and are important mediators of a variety of liver pathophysiological disorders (*Hirsova et al., 2016*; *Verma et al., 2016*). However, the role of hepatocyte exosomes in severe HS ALI awaits elucidation.

The NOD-like receptor signaling pathway is involved in the regulation of the host injury, immune and inflammatory responses (*Tannahill & O'Neill, 2011*; *Patel, 2017*). Recent studies have found that inflammatory responses mediated by the formation and activation of the NLRP3 inflammasome, an important component of NOD-like receptor

signaling, play a critical role in exacerbating severe HS liver damage (*Geng et al., 2015*). The NLRP3 inflammasome is an intracellular polyprotein complex composed of NOD-like receptor family pyrin domain-containing 3 (NLRP3), apoptosis-associated spot-like protein (ASC), and cysteinyl aspartate specific proteinase-1 (caspase-1). Activation of the NLRP3 inflammasome results in the cleavage of a large number of caspase-1 precursors (procaspase-1) into activated caspase-1 with p10 fragments, thereafter promoting the maturation and secretion of downstream proinflammatory cytokines, such as interleukin-1β (IL-1β) and IL-18, aggravating inflammation and inducing cell death.

In the current study, we analyzed the protein profile and KEGG pathways of exosomes released from hepatocytes after HS and found that NOD-like receptor signaling pathway proteins were highly enriched. We demonstrate that hepatocyte-derived exosomes can be taken up by hepatocytes and subsequently activate the NLRP3 inflammasome. These data provide new insights into the importance of hepatocyte exosomes in HS-induced ALI, and these exosomes probably act by activating the NOD-like receptor signaling pathway in hepatocytes.

## MATERIALS & METHODS

### Cell culture

HepG2 hepatocytes (human hepatocellular carcinoma cell lines, CLS Cat. # 300198/p2277_Hep-G2, RRID: CVCL_0027) were cultured in DMEM (GIBCO BRL, Life Technologies, Inc.) with 10% FBS (GIBCO BRL, Life Technologies, Inc.) at 37 °C in a 5% $CO_2$ incubator.

### HS cell model

Before HS insult, the HepG2 cell culture medium was replaced with fresh serum-free DMEM for 24 h. For the induction of HS, HepG2 cells were transferred to an incubator at a temperature of 43 °C for 1 h. Immediately after HS, the cells were cooled to 37 °C and incubated for 9 h. In some experiments, the exosome secretion inhibitor GW4869 (20 μg/ml conditioned medium; Sigma, St. Louis, MO, USA) was added to the supernatant 2 h prior to HS to inhibit the endogenous exosomes.

### Exosome isolation

The cell culture medium from control or HS HepG2 cells was collected for exosome isolation. First, cells, dead cells, and cell debris were eliminated with a series of centrifugations at 4 °C (300× g for 10 min, 2,000× g for 20 min and 10,000× g for 30 min). The supernatant was subsequently subjected to ultrafiltration through a 0.22 μm filter (Millipore, MA, USA) to remove contaminants, including apoptotic bodies, microvesicles and cell debris. The remaining supernatant was subjected to ultracentrifugation at 100,000× g for 70 min at 4 °C in a Beckman Coulter Optima TM L-80XP to pellet the exosomes. Next, the exosome pellets were washed with 1 ml of precooled sterile phosphate-buffered saline (PBS), which was again ultracentrifuged at 100,000× g for 70 min at 4 °C. Finally, the pelleted exosomes were resuspended in 30–100 μl of sterile PBS. A BCA Protein Assay kit was used to determine the protein quantity (PierceTM BCA Protein Assay kit; Thermo Fisher Scientific, MA, USA).

## Observation of exosome morphology by transmission electron microscopy (TEM)

Five microliters of an exosome suspension was added to a formvar-coated copper grid (Mecalab, QC, Canada) for 30 min, fixed in 2% paraformaldehyde for 10 min and later stained with 2% uranyl acetate for 15 min. The samples were then visualized using a Philips CM10 transmission electron microscope (JEM-2100F, Netherlands).

## Nanoparticle tracking analysis (NTA)

NTA (NS3000, Worcestershire, UK) was employed to detect the concentration and size distribution of isolated exosomes according to the manufacturer's instructions. Briefly, the exosome samples were diluted to a final concentration of 1:5,000 in sterile PBS, and each sample was analyzed three times for 60 s with NanoSight automatic analysis settings.

Comparison of the relative protein expression profiles between the control and HS-hepatocyte-derived exosomes using isobaric tags for relative and absolute quantitation technology (iTRAQ).

## Sample preparation and protein extraction

The samples were lysed with SDT [4% Sodium dodecyl sulfate (SDS) and 100 mM Tris-HCl (hydrogen chloride), pH 7.6] and mixed ultrasonically (JY96-IIN, Ningbo, China). After boiling for 15 min, the samples were centrifuged at $14,000 \times$ g for 15 min, and the precipitates were discarded. The proteins were quantified using the BCA method. Then, 20 μg of protein from each sample was added to $5 \times$ loading buffer, boiled for 5 min, and subjected to 12% SDS-PAGE (polyacrylamide gel electrophoresis). The samples were enzymatically hydrolysed by filter-aided sample preparation (FASP). Briefly, dithiothreitol (DTT) (Sigma, St. Louis, USA) was added to 30 μL of protein solution from each sample to a final concentration of 100 mM, boiled for 5 min and subsequently cooled to room temperature. The mixture was added to 200 μL of UA buffer, transferred to a 30-kD ultrafiltration centrifuge tube (Sartorius, Germany), and centrifuged at $12,500 \times$ g for 25 min. The supernatant was discarded, and 100 μL of IAA (iodoacetamide) buffer (Sigma, St. Louis, USA) [100 mM IAA in UA (uric acid) [8 M urea and 150 mM Tris-HCl, pH 8.5] was added; the mixture was allowed to react at room temperature for 30 min in the dark and then centrifuged at $12,500 \times$ g for 25 min. Next, 100 μL of UA buffer and 100 μL of $10 \times$ Dissolution buffer (AB SCIEX, USA) were added ordinally and then centrifuged at $12,500 \times$ g for 15 min after each step. The above steps were repeated twice. Finally, 40 μL of trypsin buffer [4 μg trypsin in 40 μL of Dissolution buffer] was added; the samples were shaken (GENIE Votex-2, USA) at 600 rpm for 1 min, maintained at 37 °C for 16–18 h and then centrifuged at $12,500 \times$ g for 15 min. Another 20 μL of Dissolution buffer was added, and the samples were centrifuged at $12,500 \times$ g for 15 min. The supernatant was collected, and the peptide was quantified (NanoDrop, Thermo Fisher Scientific, MA, USA).

## iTRAQ labelling

A total of 100 μg of peptide was collected from each sample and labelled with iTRAQ according to the iTRAQ Labelling Kit (AB SCIEX, USA) instructions.

### High pouvoir hydrogène (PH) reversed phase (RP) fractionation

Each set of labelled peptides was mixed and fractionated using the Agilent 1,260 Infinity II high-performance liquid chromatography (HPLC) system. The column was equilibrated with solution A (10 mM $HCOONH_4$ and 5% ACN, pH 10.0), and the sample was loaded automatically into the column and separated at a flow rate of 1 mL/min. The liquid phase gradient was as follows: 0 min–25 min, 0% solution B (10 mM $HCOONH_4$ and 85% ACN, pH 10.0); 25 min–30 min, solution B linear gradient from 0%–7%; 30 min–65 min, solution B linear gradient from 7%–40%; 65 min–70 min, solution B linear gradient from 40%–100%; and 70 min–85 min, solution B solution gradient maintained at 100%. The absorbance value at 214 nm was monitored during the elution, and the eluted fractions were collected every 1 min for a total of approximately 36 eluted fractions. The samples were lyophilized, reconstituted with 0.1% formic acid (FA) and combined into 3 parts.

### Easy n liquid chromatography (*nLC*) chromatography

Each sample was separated using a nanolitre flow rate Easy nLC system (Thermo Fisher Scientific). The column was equilibrated with 100% A solution (0.1% aqueous solution of FA), and the sample was loaded automatically and separated with an analytical column (Thermo Fisher Scientific, Acclaim PepMap RSLC 50 μm × 15 cm, nano viper, P/N 164943) at a flow rate of 300 nL/min. A 2-h liquid phase gradient was selected as follows: 0 min–5 min: solution B (0.1% aqueous solution of formic acid in acetonitrile 80% acetonitrile 6%; 5 min–105 min: a linear gradient of solution B from 6%–28%; 105 min–110 min: a linear gradient of solution B from 28%–38%; 110 min–115 min: a linear gradient of solution B from 38%–100%; and 115 min–120 min: a gradient of solution B maintained at 100%.

### Mass spectrometry

The sample was chromatographed and subjected to mass spectrometry using a Q Exactive mass spectrometer (Thermo Fisher Scientific, MA, USA). The settings were as follows: analysis time: 60 min; detection method: positive ion; mother ion scanning range: 350–1,800 m/z; primary mass spectrometer resolution: 70,000; automatic gain control (AGC) target: 3e6; and first-level Maximum IT: 50 ms. The mass/charge ratios of the polypeptide and the polypeptide fragments were collected as follows: a 10-fragment atlas was acquired after each full scan; the MS2 Activation Type was higher energy collisional dissociation (HCD); the isolation window was 2 m/z; the secondary mass spectrometer resolution was 17,500; Microscans: 1; secondary Maximum IT: 45 ms; and Normalized Collision Energy: 27 eV. The raw data for the mass spectrometry were obtained as a raw file, and the identification and quantitative analyses were performed with the Mascot 2.6 (Matrix Science) and Proteome Discoverer 2.1 (Thermo Fisher Scientific, MA, USA) software. The related parameters and descriptions are shown in Table S1.

### Kyoto encyclopedia of genes and genomes (*KEGG*) pathway annotation

In the KEGG database, KO (KEGG Orthology) is a classification system for genes and their products. Orthologous genes and their products with similar functions in the same pathway are grouped together and assigned the same KO (or K) tag. When the KEGG pathway is

annotated with the target protein cluster, the target protein sequence is KO classified by comparison with the KEGG GENES database using the KAAS (KEGG Automatic Annotation Server) software (*Tannahill & O'Neill, 2011*), and information regarding participation of the target protein sequence in the pathway is automatically obtained according to the KO classification.

### Enrichment analysis of KEGG annotations

When the target protein cluster was subjected to enrichment analysis based on KEGG pathway annotation, Fisher's exact test was applied to compare the distribution of each KEGG pathway in the target protein cluster and the overall protein cluster. Then, the significance level of protein enrichment was evaluated for a certain KEGG pathway.

### Exosome labeling and in vitro and in vivo uptake experiment

Hepatocyte derived exosomes were labeled with PKH67 Green Fluorescent Cell Linker Kit (Sigma Aldrich, St. Louis, MO, USA). Briefly, the exosomes were diluted in 1 ml of Diluent C, in parallel, 4 μl of PKH67 was added and incubated for 4 min. The staining was stopped with 2 μl of 0.5% BSA (bovine serum albumin). Ultra-centrifugation at 100,000 g for 1 h at 4 °C was performed to extract the exosomes and remove free dye. For in vitro uptake experiments, 10 μg PKH67 labeled exosomes was added into recipient HepG2 and incubated for 6 h. For in vivo uptake experiments, 40 μg PKH67 labeled exosomes was once injected into male C57/BL6 mice through tail vein. The mice were sacrificed 8 h after the injection and the liver tissue was frozen in −80 °C and sliced. The absorption of exosomes in vitro and in vivo were observed with an Olympus FV1000 confocal scanning laser microscope. The animal ethics committee of the General Hospital of Southern Theatre Command of PLA approved this study.

### Mouse HS model

Male C57/BL6 mice aged 6~8 weeks were purchased from the Guangdong Medical Laboratory Animal Centre (Guangzhou, China). The animals were acclimated for 48 h before use. To induce HS, the animals were placed in a chamber with the temperature increased to 39.0 °C within 30 min and a relative humidity of 60%. A thermocouple (Biowill, Shanghai, China) inserted 1.5 cm into the rectum was applied to measure the core temperature (Tc) at 10-min intervals. The onset of HS was considered when the Tc of the mouse reached 42.5 °C. Immediately thereafter, we removed the mice from the climate chamber and returned them to their original cages at room temperature. In some groups, GW4869 (an exosome secretion inhibitor) was given by tail injection (2 mg / kg) 2 h prior to HS ($n = 6$ per group). The animal ethics committee of the General Hospital of Southern Theatre Command of PLA approved this study.

### In Vitro and In Vivo Exosome Stimulation Experiments

In *in vitro* exosome stimulation, recipient HepG2 cells were treated with exosomes isolated from HS or Control HepG2 cells at a equal dose of 10 μg for 24 h. In *in vivo* exosome stimulation experiments, male C57/BL6 mice were infused with 40 μg exosomes isolated from HS or Control HepG2 cells trough tail vein ($n = 6$ per group). The mice were sacrificed

8 h thereafter and the liver was removed for further analysis. The animal ethics committee of the General Hospital of Southern Theatre Command of PLA approved this study.

## Western blotting

Protein extracts from exosomes or liver tissue were separated by 10% SDA-polyacrylamide gel electrophoresis (PAGE) and electrotransferred to a 0.2 μm nitrocellulose membrane. After blocking for 1 h in bovine serum albumin (BSA), the membranes were incubated with primary antibodies at 4 °C overnight and then incubated with a secondary antibody. After the membranes were washed 3 times with TBS and tween 20 (TBST), the protein bands were visualized using an Odyssey two-color infrared fluorescence imaging system (LI-COR, Nebraska, USA) according to the manufacturer's protocol and analyzed with a gel image analyzer. The antibodies used in this study were as follows: CD63 (ab10895), CD9 (ab92726, Abcam, Cambridge, UK), CD81 (18250-1-AP, Proteintech), NLRP3 (ab214185), ASC (ab47092), caspase-1 (ab1872) (Abcam, Cambridge, UK) and GAPDH (60004-1-Ig, Proteintech, Rosemont, USA).

## Alanine aminotransferase (ALT), aspartate aminotransferase (AST) and lactate dehydrogenase (LDH) assay

Cell supernatant or serum ALT, AST and LDH activity was measured using commercial ALT, AST and LDH assay kits (Nanjing Jiancheng Bioengineering Institute, Nanjing, China) according to the manufacturer's instructions.

## IL-1β assay

Cell supernatant or serum IL-1β levels were measured using a commercial ELISA kit (Meimian Industrial Co. Ltd, Jiangsu, China) according to the manufacturer's instructions.

## Histopathology

Formalin-fixed mouse liver tissues were embedded in paraffin, sectioned at 3 μm thickness and stained with hematoxylin and eosin (H&E). A fluorescence microscope Olympus Fluoview Ver.3.0 Viewer (Olympus, Tokyo, Japan) was employed to examine the histological changes.

## Immunohistochemistry (IHC)

Paraffin-embedded mouse liver tissue was sliced into 3 μm thin sections. The sections were incubated at 4 °C overnight with the following primary antibodies: MPO (22225-1-AP, Proteintech), NLRP3 (ab214185), ASC (ab47092) and caspase-1 (ab1872; Abcam, Cambridge, UK). Afterwards, the sections were incubated with peroxidase-labeled anti-rabbit (or anti-goat) secondary antibodies for 1 h at 37 °C. The expression of tissue antigens was visualized under a fluorescence microscope Olympus Fluoview Ver.3.0 Viewer (Olympus, Tokyo, Japan).

## Statistical analysis

All data are presented as the mean ± standard deviation (SD). The statistical analysis of 2 groups was performed using a two-tailed Student's t test, whereas a one-way ANOVA analysis with post hoc test was utilized to compare more than 2 groups. All experiments

were replicated three times. A value of $p < 0.05$ was considered significant. All tests were performed using GraphPad software version 7.0.

## RESULTS

### Characterization of exosomes released from hepatocytes after HS

The exosomes secreted by the control and HS-exposed HepG2 cells were isolated using the standard differential ultracentrifugation method. The morphologies were characterized by TEM. Double-layer membranous round or elliptical vesicles with diameters within 150 nm were observed under the electron microscope (Fig. 1A). Second, the size distribution of the vesicles detected by NTA was predominantly 30–150 nm (95.9 ± 2.8 nm for control exosomes vs. 108.2 ± 12.4 nm for the HS exosomes), which is mostly consistent with exosomes (Fig. 1B). Moreover, the NTA showed that HS induced a significant increase in the number of exosomes released from the HepG2 cells (Fig. 1C). Finally, we used western blotting to examine the expression of the characteristic exosomal surface markers CD9, CD63, and CD81. Figures 1D–1E shows that the CD9, CD63 and CD81 expression levels were higher in the seperated exosomes than the corresponding cell lysates, demonstrating the high level of purity of the isolated exosomes.

### Protein profiling of HS hepatocyte exosomes and KEGG pathway enrichment of the differentially expressed proteins

To explore the biological function of the HS hepatocyte exosomes, an iTRAQ-based proteomic profiling method was used to identify the differences in the protein composition of exosomes derived from control and HS hepatocytes (three independent exosome samples per group). A total of 6,677 peptides and 911 protein groups were identified. Compared to the control exosome, HS exomes showed upregulated expression of 53 (5.82%) types of proteins and downregulated expression of 21 (2.31%) other proteins. We found that several specific protein components, such as TNFSF 10, histones, S100P, HMGB1, and Diablo, were significantly enriched in HS exosomes. All of the above proteins are closely associated with cell death, injury and inflammation. Figure 2A shows the top 20 highly expressed proteins in HS exosomes. Our data indicate that HS can alter the abundance of some proteins in hepatocyte exosomes, which may be related to its biological function.

Because different proteins interact and cooperate to complete biochemical reactions, we used the KEGG database, which is commonly used in pathway research, to identify pathways that might be affected by the differential protein expression in HS exosomes. The 5 most significant pathways identified included necroptosis, the phosphatidylinositol 3-kinase (PI3K)-Akt signaling, antigen processing and presentation, apoptosis and NOD-receptor signaling pathways (Fig. 2B). Most of these pathways are associated with injury and inflammation. Therefore, we speculated that hepatocyte exosomes may participate in the development of HS-associated injury by activating the above pathways in target cells.

### In vitro and in vivo uptake of HS hepatocyte exosomes

We examined the uptake of HS hepatocyte exosomes by recipient hepatocytes through direct uptake experiments in vitro. The exosomes released from the donor HS hepatocytes

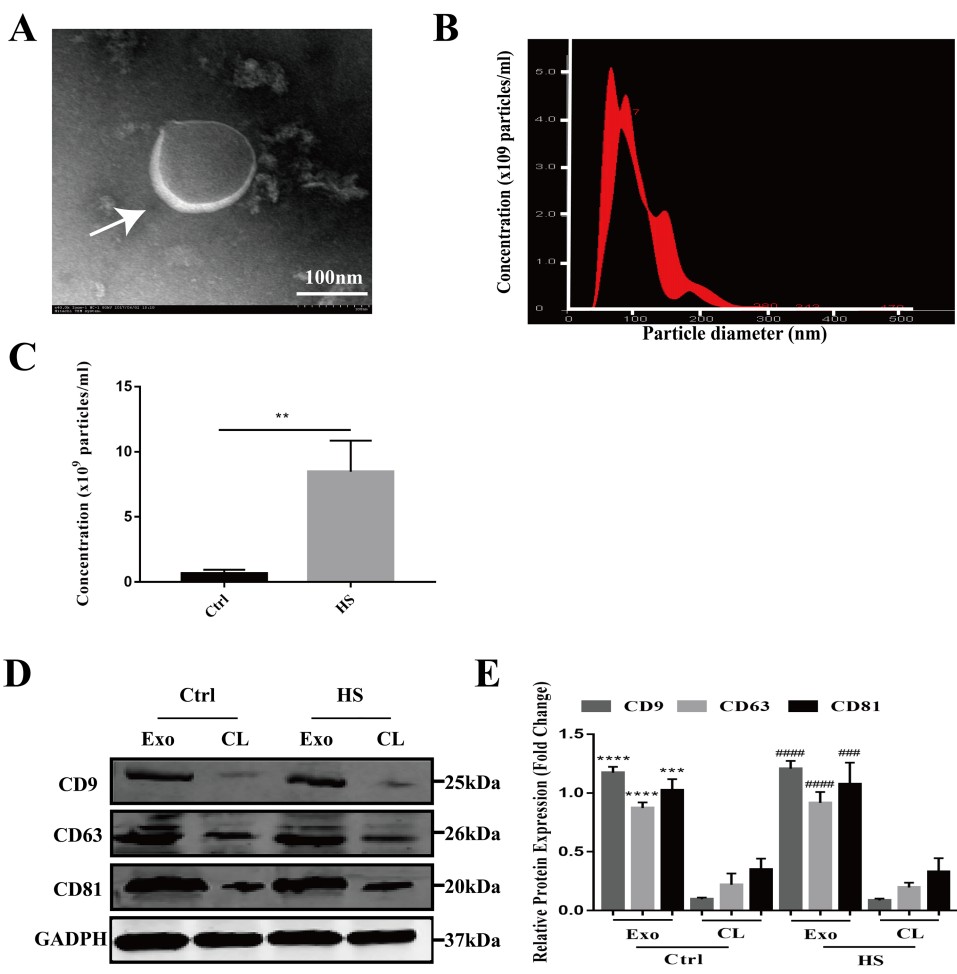

**Figure 1** **Characterization of hepatocyte exosomes.** (A) Morphology of the hepatocyte exosomes observed by TEM. The white arrow indicates the membrane bilayer around vesicles that are approximately 30-100 nm in diameter. Bar = 100 nm. (B) Size distributions of the exosomes detected by NTA showing that the diameter is predominately within the 30-100 nm range (HS hepatocyte exosome 108.2 ± 12.4 nm). (C) The concentration of control and HS hepatocyte exosomes analyzed by NTA. $^{(**)}p < 0.01$. (D) Representative bands of western blotting of exosomes (Exo) and corresponding cell lysate (CL) proteins with CD9, CD63 and CD81 antibodies. (E) Analysis of CD9, CD63 and CD81 expression relative to that of GADPH by Gel-Pro image analyzer. $^{(****)}p < 0.0001$, $^{(***)}p < 0.001$ vs. control-CL, $^{(####)}p < 0.0001$, $^{(###)}p < 0.001$ vs. HS-CL. All experiments were replicated three times.

were labeled with PKH67 and then directly added to the recipient HepG2 cells. After 6 h of incubation, confocal microscopy clearly showed that green fluorescent expression of PKH67 was evident in the HS exosome-treated hepatocytes, indicating the internalization of the exosomes by the recipient hepatocytes (Figs. 3A–3F).

The in vivo uptake of HS hepatocytes by the liver was also verified. PKH67-stained HS hepatocyte exosomes were injected into C57BL6 mice via the tail vein. Eight hours after the injection, significant absorption of exosomes by the liver was observed in frozen sections of liver tissue by confocal microscopy (Figs. 3G–3L).
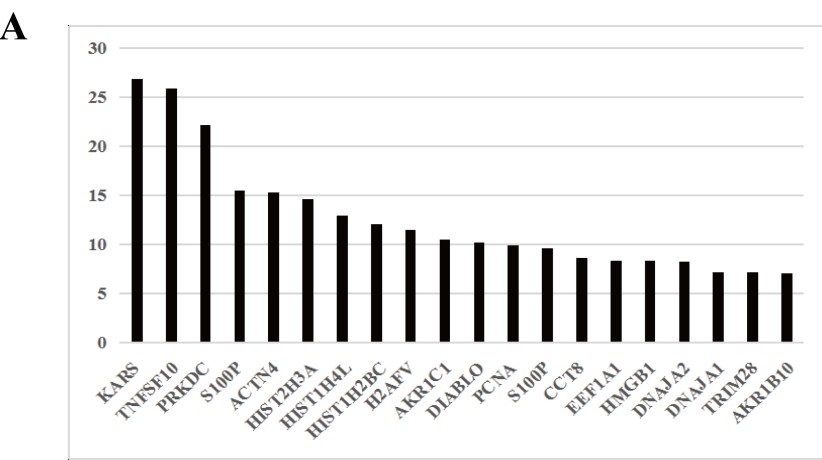

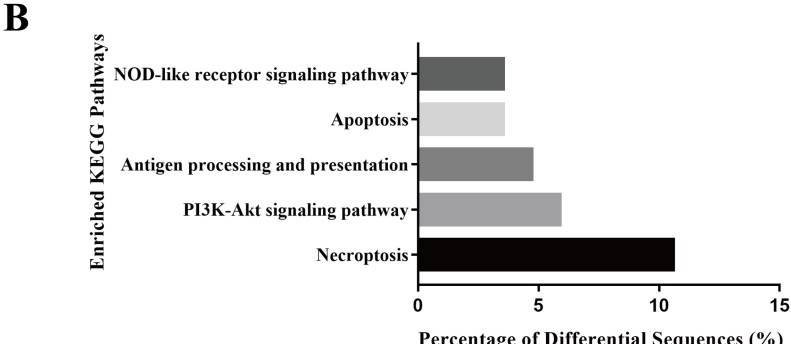

**Figure 2** **Protein profiling of HS hepatocyte exosomes and KEGG pathway enrichment of differentially expressed proteins.** (A) Identification of the differences in protein expression between exosomes derived from control and HS hepatocytes based on the iTRAQ method. Gene symbols of the 20 proteins with the most upregulated expression based on the expression fold change (HS/control) are shown. (B) The top five enriched KEGG pathways in the HS exosome proteome and the percentages of sequences involved in each pathway are shown (%).

### HS hepatocyte-derived exosomes promote NOD-like receptor signaling pathway activation and injury in recipient hepatocytes

Since bioinformatics pathway analysis of the differentially expressed proteins in the HS exosomes showed that proteins involved in the NOD-like receptor signaling pathway were enriched, we validated the NOD-like receptor signaling pathway activation properties of the HS exosomes in the recipient hepatocytes.

Recipient hepatocytes were treated with 10 μg of HS or control exosomes for 24 h. In other groups, recipient hepatocytes were subjected to HS or pretreated with GW4869 (endogenous exosome secretion inhibitor, 20 μg/ml conditioned medium) 2 h prior to HS. The expression of NOD-like receptor signaling pathway components NLRP3, ASC, and activated caspase-1 was detected in cell lysates by western blotting, and supernatant IL-1β was measured by ELISA. As shown in Figs. 4A–4C, NLRP3, ASC and activated caspase-1 expression and extracellular IL-1β levels were significantly enhanced in the recipient hepatocytes stimulated with the HS exosomes, whereas the levels in the control

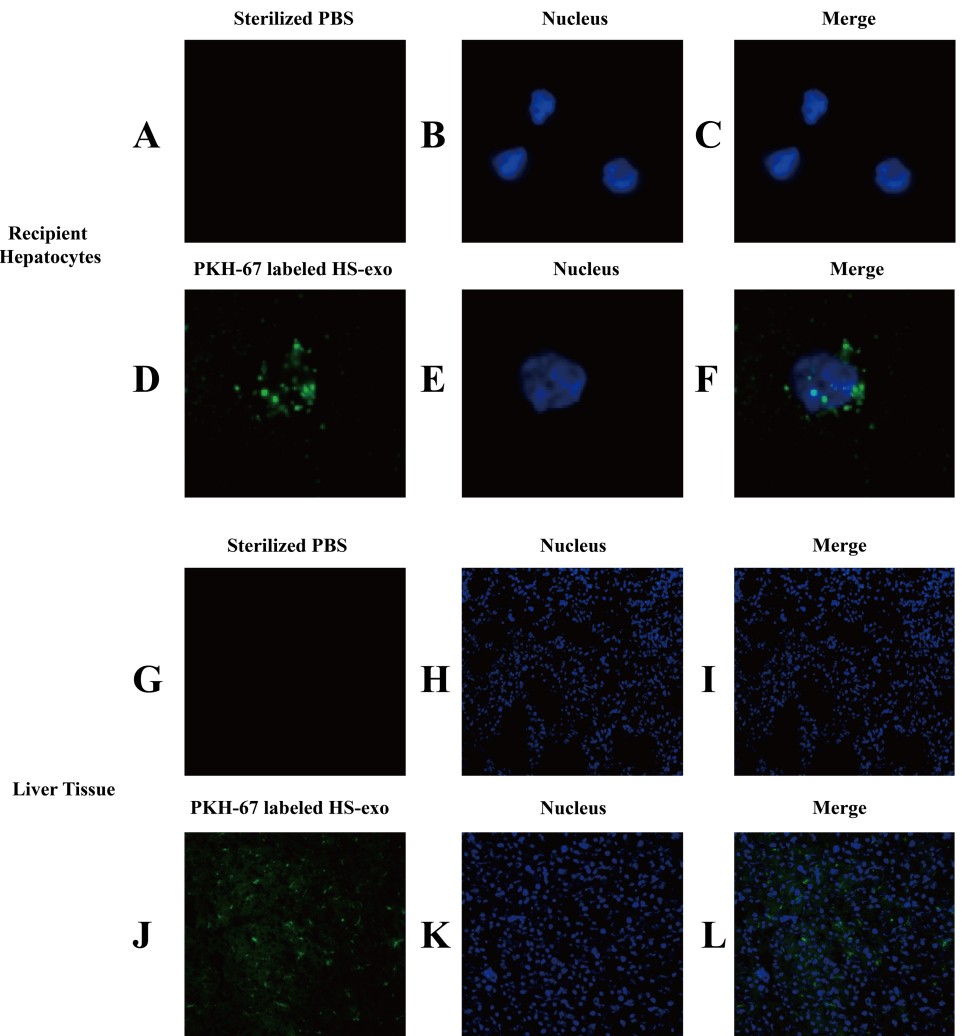

**Figure 3** **In vitro and in vivo uptake of HS hepatocyte exosomes.** (A–F) Recipient hepatocytes were incubated with PKH67-stained HS-hepatocyte exosomes or sterilized PBS for 6 h , and the internalization of the fluorescently labeled exosomes was visualized with a confocal scanning laser microscope. The control group was not treated with exosomes. (G–L) PKH67-labeled HS hepatocyte exosomes or sterilized PBS were injected into mice through the tail vein. Frozen sections of liver tissue were obtained 4 h later, and the absorption of exosomes in the liver was observed under a confocal scanning laser microscope. All experiments were replicated three times.

exosome-treated cells were similar to those of the controls. Direct HS induction significantly induced NLRP3, ASC and activated caspase-1 expression and subsequent IL-1β release, whereas pretreatment of the hepatocytes with GW4869 prior to HS significantly reduced the expression of the above components (Figs. 4A–4C). Finally, the levels of the liver markers ALT/AST/LDH in the culture supernatant were examined to evaluate hepatocyte injury. The ALT, AST and LDH levels were significantly increased in the HS exosome treatment group. The ALT/AST/LDH levels in the GW4869 pretreatment group were significantly reduced compared to those in the HS group (Fig. 4D).
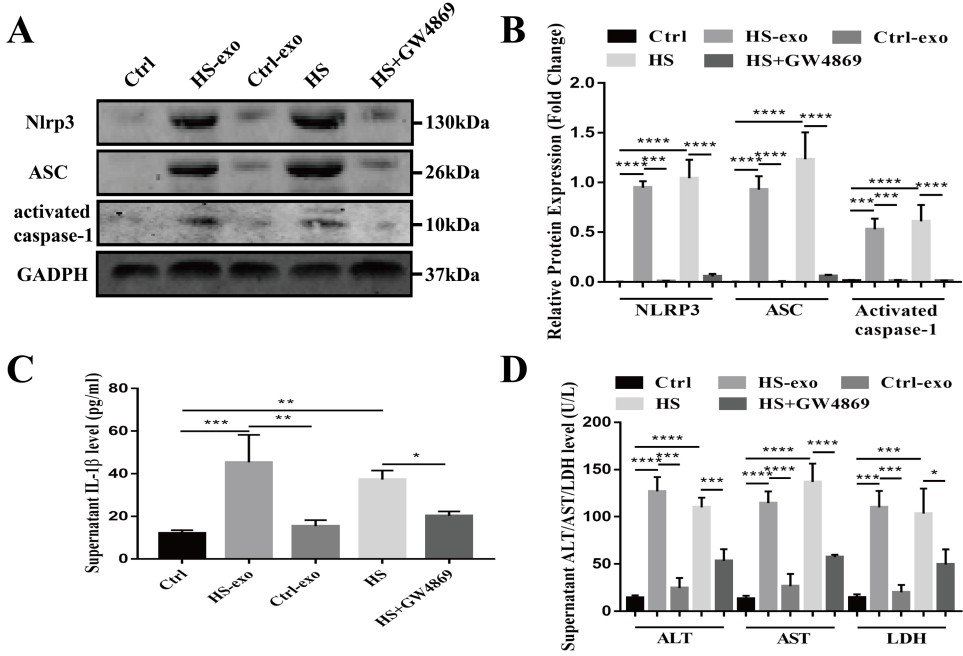

**Figure 4** **HS hepatocyte-derived exosomes promote NOD-like receptor signaling pathway activation and injury in recipient hepatocytes.** (A) The expression of NLRP3, ASC and activate caspase-1 in lysates from control hepatocytes, hepatocytes treated with control or HS exosomes, and hepatocytes subjected to HS or HS+GW4869 pretreatment assessed by western blotting. The representative bands were shown . (B) The protein expression relative to that of GAPDH by Gel-Pro image analyzer. $^{(****)}p < 0.0001$, $^{(***)}p < 0.001$. (C) Detection of IL-1 β levels in the supernatant of control hepatocytes, hepatocytes treated with control or HS exosomes, and hepatocytes subjected to HS or HS+GW4869 pretreatment, assessed by ELISA. $^{(***)}p < 0.001$, $^{(**)}p < 0.01$, $^{(*)}p < 0.05$. (D) ALT/AST/LDH levels in the supernatant of control hepatocytes, hepatocytes treated with control or HS exosomes, and hepatocytes subjected to HS or HS+GW4869 pretreatment. $^{(****)}p < 0.0001$, $^{(***)}p < 0.001$, $^{(*)}p < 0.05$. All experiments were replicated three times.

Taken together, these results show that the HS exosomes induce NOD-like receptor signaling pathway activation in recipient hepatocytes, resulting in hepatocyte injury.

## HS hepatocyte exosomes activate NOD-like receptor signaling and cause liver tissue injury in vivo

We validated the in vitro findings in vivo. Wild-type C57BL6 mice in the control or HS exosome infusion groups were injected with 40 μg of control or HS hepatocyte exosomes. In the HS or HS+GW4869 groups, hyperthermic stress was introduced in mice for 2 h before HS, and mice were injected with GW4869 (2 mg/kg). IHC and western blot was performed to detect the expression of the NOD-like receptor signaling pathway components NLRP3, ASC, and activate caspase-1 in liver tissue. As shown in Figs. 5A–5C, the positive staining and expression abundance of the above proteins was increased in the HS exosome reinfusion group compared with both the control and control exosome injection groups. The resulting release of IL-1 β into the serum following NLRP3 inflammasome activation was also elevated in the HS exosome treated group (Fig. 5D). HS promoted the activation of

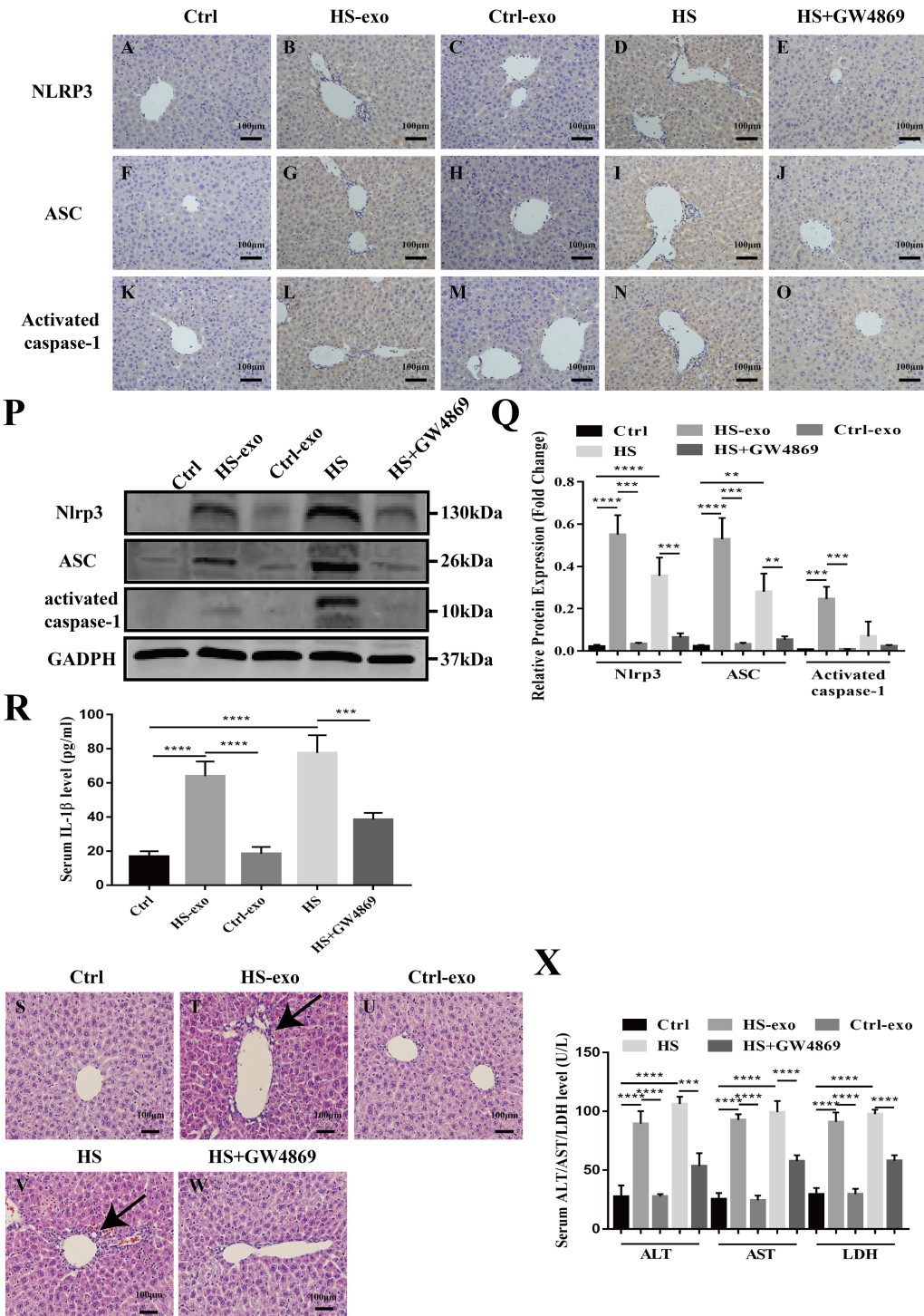

**Figure 5** **HS hepatocyte exosomes activate NOD-like receptor signaling and cause liver tissue injury in vivo.** (A–O) Expression of NLRP3, ASC and activate caspase-1 in liver tissue of control, control exosome- or HS exosome-injected, and HS or HS+GW4869 pretreatment (continued on next page...)

the NLRP3 inflammasome, while GW4869 pretreatment significantly reduced the effect of HS (Figs. 5A–5D). Furthermore, liver tissue injury was detected by H&E staining and serum ALT/AST/LDH levels. Compared to the control and control exosome-treated groups, the HS exosome infusion group displayed obvious inflammatory cell infiltration around the sinusoidal area in the liver tissue by H&E staining. GW4869 pretreatment alleviated the liver injury induced by HS (Fig. 5E). The changes in the levels of the injury markers ALT, AST and LDH in the serum presented similar trends (Fig. 5F).

## DISCUSSION

In this study, we conducted a preliminary bioinformatics analysis of exosomes released from HS-induced hepatocytes and explored the mechanisms involved in the induction of severe HS liver injury. From the results of this study, we can draw the following conclusions: (1) The morphology, diameter distribution and surface marker expression of the HS hepatocyte exosomes isolated by ultra-high-speed differential centrifugation are consistent with those of typical exosomes. (2) HS hepatocyte exosomes can be taken up by recipient hepatocytes in vitro and can also be absorbed by the liver in vivo; (3) iTRAQ protein profiling and KEGG pathway analysis shows that HS hepatocyte exosomes are rich in proteins involved in the NOD-like receptor signaling pathway; (4) HS hepatocyte exosomes can activate NOD-like receptor signaling in recipient hepatocytes in vitro and in vivo and induce liver damage and inflammation.

Exosomes are a newly discovered method of intercellular communication; they transport various signaling molecules between adjacent or distant cells, thereby participating in the functional regulation of various target cells and physiological and pathological processes (*Sato et al., 2016*). Exosome cargoes can be actively enriched in response to different stress conditions and exert different biological functions (*Javeed & Mukhopadhyay, 2017*). Therefore, the analysis of differences in the expression of their protein or genetic contents of exosomes can provide insights into the association of exosomes with disease and can be used to screen for novel biomarkers for the diagnosis and prognosis of disease. In liver disease, comprehensive proteomic and miRNA sequencing analyses have shown changes in serum exosomal protein/miRNA expression profiles in drug-induced liver injury (DILI) and alcoholic hepatitis (*Baker et al., 2015*; *Momen-Heravi et al., 2015*; *Povero et al., 2014*). In nonalcoholic hepatitis, the association between highly expressed proteins in exosomes and their pathological mechanisms has also been revealed, and candidate

proteins/miRNAs have been selected as novel markers for predicting organ damage (*Götz et al., 2008*). Hepatocytes are the main parenchymal and exosome-secreting cells in the liver. Hepatocytes are vulnerable to thermal injury and actively participate in the induction of liver damage. We explored how HS hepatocyte-derived exosomes mediate hepatocyte communication and their functional roles in liver injury to provide new insight into the pathogenesis of HS liver injury.

The mechanism of HS liver injury is still unclear. The liver pathology in HS animal models is mainly characterized by abundant neutrophil infiltration, sinusoidal endothelium denudation, local microthrombus formation in the hepatic sinus, hepatocyte degeneration and membrane ballooning or rupture, which is particularly obvious in the sinusoidal area. These findings suggest that heat may cause direct damage to liver cells, but the multiple types of secondary damage initiated after hepatocyte injury, such as an unbalanced inflammatory response, may be more important than the initial injury for the hepatocyte "double hit." In our study, the KEGG pathway analysis showed that the highly expressed proteins in exosomes released in response to HS stimulation were closely associated with various inflammatory signaling pathways, such as the NOD-like receptor signaling pathway, necroptosis (a type of inflammatory cell death), and antigen presentation and processing. Exosomes released from HS hepatocytes may play a key role in the induction of hepatic inflammation. The NLRP3 inflammasome is a major intracellular polyprotein complex that induces an inflammatory response by mediating immune cell infiltration and organ damage. Studies have shown that GalN/LPS induces enhanced protein expression of the NLRP3 inflammasome components, including NLRP3, ASC and caspase-1 (*Seo et al., 2017*; *Zhang et al., 2017*; *Chen et al., 2016*; *Liu et al., 2016*). Geng Yan, et al. also confirmed that HS-injured hepatocytes release the endogenous signaling molecule high mobility protein-1 (HMGB1), which activates the hepatocyte NLRP3 inflammasome, leading to pyroptosis and further expansion of the inflammatory response and liver damage (*Geng et al., 2015*).

In this study, we found that HS hepatocyte exosomes activate the NLRP3 inflammasome of hepatocytes in vitro, activate NLRP3 in the liver in vivo and induce liver damage. Therefore, we propose that HS exosomes promote liver damage and inflammation, possibly through the activation of NOD-like receptor signaling, but we did not directly confirm the causal relationship between the two. In addition, HS hepatocyte exosomes contain a variety of important proteins involved in NOD-like receptor signaling pathways, which are closely related to inflammatory responses. However, in this study, we did not conduct experiments to directly verify that these proteins mediate the proinflammatory functions of exosomes, and further confirmation is needed in future studies. These proteins may be targets for the treatment of HS liver damage. In addition, other inflammatory signaling pathways in target cells may also be activated and involved in the inflammation process. Due to the diverse exosome protein composition, these results highlight the complexity of exosomes. In addition to containing proteins, exosomes contain other components, such as nucleic acids and lipids, which also affect the biological functions of exosome-target cells and deserve further study. In addition, our results did not fully reflect the effect of HS hepatocytes on liver damage in vivo because we chose hepatocyte cell lines as exosome

donor cells rather than primary hepatocytes. Data on primary hepatocyte exosomes will be more convincing than data on cell line exosomes.

## CONCLUSIONS

In conclusion, this study preliminarily confirms that exosomes released by HS-induced hepatocytes are involved in HS ALI and may regulate the activation of the NLRP3 inflammasome pathway. This study opens up a new perspective into the function of hepatocyte exosomes in severe HS, and provides a basis for the subsequent verification of the biological functions of NOD-like pathway-associated proteins in HS hepatocyte exosomes with the hope of discovering new intervention targets.

## ACKNOWLEDGEMENTS

The authors are grateful to Mr. Panweilun for the technical assistance of NTA. We thank the Guangzhou Branch of Chinese Academy of Life Sciences for technical assistance of TEM.

### Funding

This work was supported by the National Natural Science Foundation of China (grant no. 81671896), the Military Medical Innovation Project (grant No. 18CXZ032), the Natural Science Foundation of Guangdong Province (grant No. 2019A1515012088) and the Guangdong Provincial Science and Technology Plan Project (grant No. 2017A020215055). The funders had no role in study design, data collection and analysis, decision to publish, or preparation of the manuscript.

### Grant Disclosures

The following grant information was disclosed by the authors:
National Natural Science Foundation of China: 81671896.
Military Medical Innovation Project: 18CXZ032.
Natural Science Foundation of Guangdong Province: 2019A1515012088.
Guangdong Provincial Science and Technology Plan Project: 2017A020215055.

### Competing Interests

The authors declare there are no competing interests.

### Author Contributions

- Yue Li conceived and designed the experiments, performed the experiments, analyzed the data, contributed reagents/materials/analysis tools, prepared figures and/or tables, authored or reviewed drafts of the paper, approved the final draft.
- Xintao Zhu performed the experiments, analyzed the data, contributed reagents/materials/analysis tools, prepared figures and/or tables, approved the final draft.

- Ming Zhang performed the experiments, contributed reagents/materials/analysis tools, prepared figures and/or tables, authored or reviewed drafts of the paper, approved the final draft.
- Huasheng Tong conceived and designed the experiments, performed the experiments, contributed reagents/materials/analysis tools, authored or reviewed drafts of the paper, approved the final draft.
- Lei Su conceived and designed the experiments, authored or reviewed drafts of the paper, approved the final draft.

## Animal Ethics

The following information was supplied relating to ethical approvals (i.e., approving body and any reference numbers):

The animal ethics committee of the General Hospital of Southern Theatre Command of PLA approved this study.

## Data Availability

The raw measurements are available in the Supplemental Files.

## Supplemental Information

Supplemental information for this article can be found online at http://dx.doi.org/10.7717/peerj.8216#supplemental-information.

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
