# Peer review of "Heatstroke-induced hepatocyte exosomes promote liver injury by activating the NOD-like receptor signaling pathway in mice"

_PeerJ, doi:10.7717/peerj.8216_

## Round 0.1 · original submission · Major Revisions

The reviewers have reported some major concerns, especially from the second reviewer. English writing needs to improve.

Reviewer 1 ·

Basic reporting

First, the manuscript contains numerous counts of unclear, ambiguous, and unscientific English writing, which is reflected by inconsistent, misleading, exaggerating, and/or inaccurate statements as well as grammatic errors and misspellings.

Examples include but not limited to:
Line 223 describes “Fig.2A showed the top 10 high-expressed proteins..” whereas in the corresponding figure (erroneously Fig4), top 20 most abundant proteins were shown.
HS is defined as “heat stress” in the abstract (line 42) but “heatstroke” in the introduction (line 71).
In line 217, does “three exosomes (should be “exosome”) samples for each group” mean three independent experiments or three technical repeats in one experiment?
In line 192, authors stated that “The trafficking of exosomes in vitro and in vivo were observed with ….” However, the static images of fluorescence staining cannot indicate the movement, i.e. trafficking, of the exosomes.

Second, the introduction fails to provide sufficient background to validate the rational of the study due to inaccurate citation of references. None of the first 5 references cited in the first two paragraphs of the introduction support the statements. For example, authors stated “A multicenter epidemiological 78 survey of inpatients showed that approximately 31.9% of HS patients developed ALI [3],” in lines 77-78, when Reference#3 is a case report of a single human subject.

Third, the figures are in a different order than described in the text. Figure 4 and 5 in the figures section should be Figure 2 and 3 based on the text.

Experimental design

Authors intended to investigate whether exosomes are induced by heatstroke and then contributed to heatstroke-induced liver injury by combining in vitro and in vivo assays. The strategy described in the manuscript add strengths to the study; however, the method section largely lacks important and necessary details and information to confirm the validity of the results or to replicate. Major points include:
1. iTRAQ based proteomic profiling and KEGG pathway enrichment were major analyses used to explore differential protein expression in the exosomes caused by HS; these procedures should be described in the method with respect to the experiment protocol, the definition of “kinds/groups of proteins” and significance.
2. How many mice were used in each group of the in vivo study? Had the animals been acclimated before use? Were the experiments reviewed and approved by the Instructional Animal Care and Use Committee (IACUC)?
3. Many commercial kits (eg, “a Odyssey kit”) and equipment (eg, “a fluorence microscope”) lack specific name or model description.

Validity of the findings

The validity of the results cannot be verified, mainly due to a) the lack of detailed description and information for the experimental procedures; and b) unclear and ambiguous description of the results.

Additional comments

Authors should improve the overall writing of the manuscript on both the scientific and English aspects to ensure accuracy and clarity.

Reviewer 2 ·

Basic reporting

The figures are out of order. Otherwise, the writing is clear and effective, but could use grammar check.

Experimental design

Overall, the study is design and execution are acceptable/

Validity of the findings

The data are sound and the conclusions are mostly supported by the data, though I'm concerned about possible mix-ups in samples or otherwise poor labeling of their data.

Additional comments

Here, the authors tested a novel hypothesis in an understudied field (heat stress-induced liver injury). The experimental design is adequate. However, there are several issues to address:
Major issues...
1) All figures: The figures are out of order.
2) Current figure 2 (hepatocytes treated with exosomes): The blots and the densitometry in panel A do not match. The blots shower lower abundance of inflammasome markers with HS-exo treatment. The authors need to be certain their data are correct and they know which sample is which on the blot, and then correct their labeling.
3) Current figure 3 (immunohistochemistry for inflammasome markers): The "positive" staining just looks like diffuse background staining to me. The authors should support these data with western blots.
4) Current figure 5 (uptake of exosomes): It appears that the authors tested uptake of exosomes by hepatocytes in control mice. What about in mice with hyperthermia, which is what they are really interested in? It is entirely possible that uptake differs in hyperthermic mice.
Minor issues...
1) Overall, the manuscript is well-written, but the English grammar could use some minor improvement.

---

## Round 0.2 · Minor Revisions

Please solve the comments that the reviewer proposed regarding additional references in the introduction.

Reviewer 2 ·

Basic reporting

No additional comments.

Experimental design

No additional comments.

Validity of the findings

No additional comments.

Additional comments

It took me considerable time to locate new reference 3 due to the obscurity of the journal (Med J Chin PLA). I understand that the authors cannot control where this paper was published, but it would be preferable if the authors could provide more accessible references in the early Introduction.

---

## Round 0.3 · accepted · Accept

Please provide more detail to help locate new reference 3 as suggested by the reviewer in the references, during the proof stage